# Naphthalene-Based Oxime Esters as Type I Photoinitiators for Free Radical Photopolymerization

**DOI:** 10.3390/polym14235261

**Published:** 2022-12-02

**Authors:** Zhong-Han Lee, Shih-Chieh Yen, Fatima Hammoud, Akram Hijazi, Bernadette Graff, Jacques Lalevée, Yung-Chung Chen

**Affiliations:** 1Department of Chemical and Materials Engineering, National Kaohsiung University of Science and Technology, Kaohsiung 80778, Taiwan; 2Université de Haute-Alsace, CNRS, IS2M UMR 7361, F-68100 Mulhouse, France; 3Université de Strasbourg, F-67200 Strasbourg, France; 4Plateforme de Recherche et D’analyse en Sciences de L’environnement (EDST-PRASE), Beirut P.O. Box 6573/14, Lebanon; 5Photo-SMART (Photo-Sensitive Material Advanced Research and Technology Center), National Kaohsiung University of Science and Technology, Kaohsiung 80778, Taiwan

**Keywords:** free radical photopolymerization, methoxy, naphthalene, oxime ester, substitution, Type I photoinitiator

## Abstract

In order to discuss the polymerization effect from the substituted position and methoxy group of Type I photinitiators, a series of naphthalene-based oxime esters was designed and synthesized. Compared to the 2-naphthalene-substituted compound, the UV absorption region of the 1-naphthalene-based compound was greatly improved. In addition, the methoxy substitution exhibited longer absorption characteristics than did the methoxy-free one. The photochemical reaction behavior of these novel compounds was also studied by photolysis, cyclic voltammetry (CV), and electron spin resonance (ESR) experiments. Finally, the initiation abilities of naphthalene-based oxime esters toward trimethylolpropane triacrylate (TMPTA) monomer were conducted through the photo-DSC instrument under UV and a 405@nm LED lamp. Remarkedly, the naphthalene-based oxime ester (**NA-3**) that contains 1-naphthalene with *o*-methoxy substituent showed the rather red-shifted absorption region with the highest final conversion efficiency under UV (46%) and 405@nm LED (41%) lamp irradiation.

## 1. Introduction

Photopolymerization, compared to the tradition thermal polymerization, exhibited advantages such as low energy consumption, faster curing reaction times, no presence of volatile organic compounds, and low cost [1,2]. In addition, the photopolymerization kinetics can be precisely controlled spatially and temporally compared to the thermal polymerization [3,4,5]. Furthermore, both of the living and nonliving radical photopolymerization had been adopted and developed currently. For example, a photoinduced methodology with controlled radical polymerizations allowed the precise control of molecular weight (MW) and molecular weight distribution [6,7,8]. On the other hand, the nonliving radical polymerization is most widely used in various established and emerging applications, including coatings, inks, dentals, 3D printings, and microelectronics [9,10,11,12,13,14]. This kind of polymerization photopolymerization is generally classified into two pathways, (1) the free radical polymerization (FRP) [15] and (2) the cationic polymerization (CP) [16]. Among them, FRP had been utilized in an acrylate-based system and many of them were adopted in coatings, adhesions, and photolithography industrial applications [15,17].

Based on the radical generation mechanism for the nonliving types, there are two types for FRP systems including (1) a Type I (one-component photoinitiator) and (2) a Type II photoinitiaing system (two-components photoinitiator requiring the hydrogen acceptor and hydrogen donor package). Among all, the Type I photoinitiator is more efficiently on the radicals’ generation. In addition, this one-component photoinitiator can avoid the phenomenon of uneven mixability in the phototinitiating system [18]. Aromatic organic compounds such as ketals, benzoins, acetophenones, oxime esters, acylphosphine oxides, etc., are considered as the common structural backbone skeleton for the Type I photoinitiators [19,20,21]. Oxime esters are currently under intense investigation because of their highly efficient N–O bond cleavage propensity. That is, the N–O bond is broken easily from the light irradiation and then generated the iminyl and carbonate radicals. Furthermore, the decarboxylation process-generated CO_2_ might solve the polymerization inhibition in the presence of oxygen [22,23] (shown below). Up to now, some heterocyclic-based oxime esters such as coumarin-based [24,25], carbazole-based [26,27,28,29], phenothiazine-based [30], thiophene-based [31,32], and chalcones-based photoinitiators [33], were designed for achieving higher photoreactivity under specific light irradiation conditions. In our previous study, we also synthesized series of triphenylamine-based oxime esters [18,34,35]. Obviously, through the modification, various oxime esters had different characteristics and reactivities.

Currently, fused aromatic rings were also adopted and prepared in oxime ester systems. Lalevee et al. developed two series of pyrene and anthracene-based oxime esters [36]. The results showed that pyrene-based oxime esters were the most sufficient photoinitiating systems compared to anthracene ones. Alternative fused aromatic rings, naphthalene and its derivatives have also been designed to be used as efficient photoinitiators [37]. A naphthalene-based structure is expected to have a planar geometry with good electron delocalization characteristics. Therefore, in this work, a series of naphthalene-based oxime esters (**NA-1–4**) were prepared (Figure 1). Through the changing of the substituted position and the methoxy substituent of the naphthalene moiety, their photochemical properties and photoinitiating abilities are investigated and compared in detail. The naphthalene chromophore is assumed to react mainly through a triplet state pathway (triplet state (T_1)_ quantum yield ~80% vs. singlet state (S_1_) quantum yield ~20%) [38]. In addition, the presence of a methoxy group (e.g., for 1-methoxy naphthalene) significantly decreases the intersystem crossing (from ~80% to ~50%) and increases the singlet state quantum yield from ~20% to ~40%) [39]. The study of new oxime esters based on naphthalene chromophore is very interesting to shed some light on the S_1_ or T_1_ cleavage processes.

## 2. Experimental

### 2.1. Materials

2-Naphthaldehyde (98%, Sigma-Aldrich, St. Louis, MO, USA), 1-naphthaldehyde (95%, Alfa Aesar, USA), 1-methoxynaphthalene (98%, Sigma-Aldrich, USA), 2-methoxynaphthalene (98%, TCI, Japan), phosphorus oxychloride (POCl_3_) (99.9%, Sigma-Aldrich, USA), acetic anhydride (99.5%, Sigma-Aldrich, USA), anhydrous sodium acetate (SHOWA, Taiwan), hydroxylamine hydrochloride (99%, ACROS ORGANICS, USA), triethylamine (5 vol.%, FERAK, USA), trimethylolpropane triacrylate (TMPTA) (99%, DuPont, Taiwan), N-*tert*-butyl-α-phenylnitrone (PBN) (98%, Alfa Aesar, USA) and all other chemical reagents, unless otherwise specified, were used directly without purification. Solvents were distilled over appropriate drying agents and used or stored under a N_2_ atmosphere.

### 2.2. Synthesis of Aldehyde Precursors and Target Photoinitiators

Two aldehyde precursors (2-methoxy naphthaldehyde and 4-methoxy-1-naphthaldehyde) were synthesized by the Vilsmeier–Haack reaction as described in the following.

#### 2.2.1. 2-Methoxy Naphthaldehyde

POCl_3_ (3.55 mL, 37.7 mmol) was added dropwise at 0 °C in DMF (20 mL) under N_2_. The reactive mixture was stirred for 30 min and then 2-methoxynaphthalene (5.416 g, 34.2 mmol) was added and heated to 95 °C for 4 h. After cooling to the room temperature, the solution was neutralized with sodium acetate solution (2M). The mixture was extracted with DCM, and the organic extract was collected and dried over anhydrous MgSO_4_. Finally, the crude residue was purified by silica gel chromatography using DCM: n-Hexane = 1:1 as an eluent to afford a light-yellow powder compound (41%, 2.61g). ^1^H NMR (400 MHz, CDCl_3_, δ, ppm): 10.771 (s, 1H, -CHO), 9.118–9.096 (d, 1H, *J* = 8.8 Hz, Ar-H), 8.299–8.276 (d, 1H, *J* = 9.2 Hz, Ar-H), 7.950–7.930 (d, 1H, *J* = 8 Hz, Ar-H), 7.657–7.586 (2H, Ar-H), 7.476–7.436 (1H, Ar-H), 4.059 (s, 3H, -OCH_3_).

#### 2.2.2. 4-Methoxy-1-Naphthaldehyde

4-Methoxy-1-naphthaldehyde (light-yellow powder compound, yield = 55%) was synthesized from 1-methoxynaphthalene as a starting material and similar procedures to that used to obtain 2-methoxy naphthaldehyde mentioned above were used. ^1^H NMR (400 MHz, CDCl_3_, δ, ppm): 10.172 (s, 1H, -CHO), 9.312–9.290 (d, 1H, *J* = 8.8 Hz, Ar-H), 8.313–8.292 (d, 1H, *J* = 8.4 Hz, Ar-H), 7.862–7.842 (d, 1H, *J* = 8 Hz, Ar-H), 7.702–7.660 (1H, Ar-H), 7.574–7.533 (1H, Ar-H), 6.855–6.835 (d, 1H, *J* = 8 Hz, Ar-H), 4.046 (s, 3H, -OCH_3_).

#### 2.2.3. **NA-1–4** Photoinitiators

Taking **NA-1** as an example. 2-Naphthaldehyde (3 g, 19.2 mmol), anhydrous sodium acetate (3.15 g, 38.4 mmol), and hydroxylamine hydrochloride (2.7 g, 38.4 mmol) were dissolved in a EtOH (50 mL) solvent. The solution was stirred at reflux temperature for 2 h under N_2_ atmosphere. Once the reaction was complete, the solvent was evaporated to get the unpurified product. This crude product was then dissolved in CHCl_3_ (20 mL) in a round-bottomed flask directly. Then, triethylamine (3.25 mL, 23.4 mmol) was introduced into the solution. Finally, a dropwise of acetic anhydride (2.22 mL, 23.4 mmol) was added. The solution was stirred at reflux temperature for 2 h. After cooling to the room temperature, the mixture was extracted using CH_2_Cl_2_/H_2_O, and the organic extract was collected and dried over anhydrous MgSO_4_. The crude product was purified through silica gel chromatography (using a DCM/n-hexane = 1/1 eluent) to afford a target light-yellow powder (59%, 2.4 g). ^1^H NMR (400 MHz, CDCl_3_, δ, ppm):8.506 (s, 1H, N=C-H), 8.046 (s, 1H, Ar-H), 7.990–7.852 (4H, Ar-H), 7.587–7.514 (2H, Ar-H), 2.269 (s, 3H, -CH_3_). ^13^C NMR (100 MH_Z_, CDCl_3_, δ, ppm):168.83, 155.99, 150.41, 134.88, 134.13, 134.07, 133.10, 132.85, 130.70, 129.70, 129.15, 128.99, 128.84, 128.59, 128.57, 128.48, 128.37, 128.27, 128.01, 127.91, 127.80, 127.68, 127.61, 126.93, 126.83, 126.58, 126.30, 123.26, 122.70, 19.61. FT-Mass (*m*/*z*) found 236.06810 (M-Na^+^). EA (C_13_H_11_NO_2_): calcd. C, 73.23; H, 5.2; N, 6.57; found C, 71.67; H, 5.17; N, 6.36%.

**NA-2** (light-yellow powder, yield = 55%). ^1^H NMR (400 MHz, CDCl_3_, δ, ppm):8.994 (s, 1H, N=C-H), 8.595–8.573 (d, 1H, *J* = 8.8 Hz, Ar-H), 7.996–7.975 (d, 1H, *J* = 8.4 Hz, Ar-H), 7.925–7.906 (d, 1H, *J* = 7.6 Hz, Ar-H), 7.665–7.508 (3H, Ar-H), 2.300 (s, 3H, -CH_3_). ^13^C NMR (100 MH_Z_, CDCl_3_, δ, ppm):168.77, 155.75, 133.69, 132.33, 130.75, 129.51, 128.84, 127.80, 126.45, 126.00, 125.11, 124.39, 19.65. FT-Mass (*m*/*z*) found 236.06807 (M-Na^+^). EA (C_13_H_11_NO_2_): calcd. C, 73.23; H, 5.2; N, 6.57; found C, 73.11; H, 5.08; N, 6.56%.

**NA-3** (yellow powder, yield = 53%). ^1^H NMR (400 MHz, CDCl_3_, δ, ppm):9.146 (s, 1H, N=C-H), 9.039–9.017 (d, 1H, *J* = 8.8 Hz, Ar-H), 7.960–7.938 (d, 1H, *J* = 8.8 Hz, Ar-H), 7.799–7.777 (d, 1H, *J* = 8.8 Hz, Ar-H), 7.640–7.597 (1H, Ar-H), 7.432–7.392 (1H, Ar-H), 7.268–7.245 (1H, Ar-H), 3.997 (s, 3H, -OCH_3_), 2.295 (s, 3H, -CH_3_). ^13^C NMR (100 MH_Z_, CDCl_3_, δ, ppm):169.09, 158.48, 153.42, 134.95, 134.20, 131.99, 129.09, 128.87, 128.39, 128.34, 128.28, 127.74, 126.14, 124.98, 124.34, 123.94, 112.60, 112.12, 111.92, 56.50, 56.46, 19.75. FT-Mass (*m*/*z*) found 266.07860 (M-Na^+^). EA (C_14_H_13_NO_3_): calcd. C, 69.12; H, 5.39; N, 5.76; found C, 69.35; H, 5.33; N, 5.66%.

**NA-4** (yellow powder, yield = 39%). ^1^H NMR (400 MHz, CDCl_3_, δ, ppm):8.848 (s, 1H, N=C-H), 8.668–8.647 (d, 1H, *J* = 8.4 Hz, Ar-H), 8.353–8.332 (d, 1H, *J* = 8.4 Hz, Ar-H), 7.822–7.801 (d, 1H, *J* = 8.4 Hz, Ar-H), 7.676–7.634 (1H, Ar-H), 7.570–7.529 (1H, Ar-H), 6.852–6.832 (d, 1H, *J* = 8 Hz, Ar-H), 4.051 (s, 3H, -OCH_3_), 2.282 (s, 3H, -CH_3_). ^13^C NMR (100 MH_Z_, CDCl_3_, δ, ppm):168.95, 158.57, 156.09, 131.70, 131.56, 128.34, 125.78, 125.65, 124.60, 122.60, 118.40, 55.68, 19.68. FT-Mass (*m*/*z*) found 266.07861 (M-Na^+^). EA (C_14_H_13_NO_3_): calcd. C, 69.12; H, 5.39; N, 5.76; found C, 69.18; H, 5.58; N, 5.73%.

### 2.3. Measurement

The proton-nuclear magnetic resonance (^1^H NMR) and carbon-nuclear magnetic resonance (^13^C NMR) spectra of the samples were conducted under room temperature by using an Agilent Unity plus-400 spectrometer. All the testing samples were dissolved in deuterated chloroform. Fourier-transform mass spectrometry (FT-Mass) was recorded using a JEOL AccuTOF GCx-plus instrument, Japan. Elemental analyzer (EA) was performed on an elemental analyzer equipment (elementar, model: UNICUBE, Germany). Ultraviolet–visible (UV-vis) and photoluminescence (PL) spectra of the samples were dissolved in dichloromethane (DCM) (concentration = 1 × 10^−5^ M) and measured by using a PerkinElmer Lambda 35 UV–visible and Hitachi F-4500 spectrometer, US (excitation wavelength = 330 nm), respectively. Cyclic voltammetry (CV) of the samples were measured by using a BioLogic SP-150 model at a scan rate of 100 mV s^−1^ in the range of 0 to −2 V and 0 to 2.0 V. All testing was conducted at room temperature in DCM solution (concentration = 1 × 10^−3^ M) with a conventional three-electrode configuration, in which the three electrodes were a platinum working electrode, a platinum wire auxiliary electrode, and an Ag/Ag^+^ reference electrode. Melting point (T_m_) of the samples was determined under nitrogen atmosphere with a scan rate of 10 °C min^−1^ by using Perkin Elmer DSC 6000A. Thermal decomposition temperature (T_d_) of the samples was conducted under nitrogen atmosphere with a heating rate of 15 °C min^−1^ by using the TA Instruments SDT Q600 Simultaneous DTA-TGA instrument. The samples analyzed had a mass of 3–5 mg and the Td was taken as the temperature at which 5% weight loss had occurred. Photolysis result of the selected samples (**NA-1** and **NA-2**) was dissolved in DCM (concentration = 1 × 10^−5^ M) and measured by using a PerkinElmer Lambda 35 UV–visible. The absorption intensity of the solution was determined at various durations of exposure under a Philips 16 W lamp (type Actinic BL; λ = 365 nm). Electron spin resonance (ESR) spectroscopy of the selected samples was conducted at room temperature under nitrogen atmosphere by using a Bruker EMX Plus X-Band spectrometer. Ultra-high-pressure mercury (MUV-250U-L, λ = 250–450 nm, microwave power = 5 mW) was used as an irradiation source to generate related radicals. The radicals were trapped by N-*tert*-Butyl-α-phenylnitrone (PBN), and tert-butylbenzene as a solvent in accordance with a procedure described in the literature [40]. The concentration of PBN and oxime esters was 1 × 10^−2^ and 1 × 10^−3^ M, respectively.

The photopolymerization reactivity of the formulations was measured under nitrogen atmosphere (flow rate = 20 mL min^−1^) at 30 °C by using Perkin Elmer DSC 6000 photo-DSC analysis instrument, US. A UV lamp (intensity = 180 mW cm^−2^; λ = 250–450 nm) or LED@405 nm (intensity = 180 mW cm^−2^) was used as radiation sources. The composition was containing oxime esters (2 wt%), and TMPTA (98 wt%) without any additional solvent. Approximately 15 mg of a composition was placed in an aluminum DSC pan, which was used to ensure the formulations had similar thickness. Heat flow versus time curves were recorded to analyze the reaction of unsaturated moieties in the system during light irradiation. By integrating the area under the exothermic peak, the double bond conversion efficiency (DC; %) could be calculated using the following Equation (1) [41]:DC = (ΔH_t_/n × ΔH_o_^theor^) × 100%(1)
where ΔH_t_ is the total reaction heat enthalpy within the exposure time, and ΔH_o_^theor^ is the theoretical reaction heat enthalpy of one acrylate for complete conversion. ΔH_o_^theor^ was 86 kJ mol^−1^ [42]. In addition, n is number of acrylates per monomer molecule.

In addition, the rate of polymerization (R_p_) is directly related to the heat flow (dH/dt) by Equation (2) [43]:R_p_ = dC/dt(2)

## 3. Results and Discussion

### 3.1. Synthesis and Photophysical Properties

To investigate the photoreactivity of the methoxy and position-substituted naphthalene-based oxime esters, in this study, four oxime esters (**NA-1–4**) were obtained using several facile steps: (a) the Vilsmeier–Haack formylation reaction involving aromatic napthalenes, POCl_3_, and DMF to generate the corresponding aldehyde precursors; (b) the oximation reaction between carbonyl compound with hydroxylamine hydrochloride, and (c) the condensation reaction with acetic anhydride in the presence of a base as a catalyst (Figure 1). Remarkedly, the intermediates in the oximation reaction step are directly subjected to a condensation reaction without purification. The oxime esters could be obtained with reaction yields in the range of 39 to 59%. All the oxime esters were characterized through ^1^H NMR, ^13^C NMR and high-resolution mass spectroscopy (HRMS) (Appendix A). In addition, the elemental analysis (EA) (see experimental part) of the oxime esters is also conducted. All the spectra and data are well consistent with the purposed ones demonstrating that the oxime esters have been synthesized successfully. In addition, all the oxime esters showed good dispersity in the testing monomer TMPTA. Therefore, a good mixability was expected to be achieved for the photoinitiating compositions.

UV–visible absorption spectra of the oxime esters (**NA-1–4**) were conducted in DCM and the related curves and data are illustrated in Figure 1a and Table 1. The spectra of this series contained a board absorption region below 400 nm. The absorption peaks were attributed to the π–π* electronic transition accompanied by slight charge-transfer characteristics. The maximum absorption in the longer-wavelength band for **NA-1–4** was 294, 312, 348, and 328 nm, respectively. **NA-2** had longer absorption region than did the **NA-1** compound because the former compound had a longer conjugation length. In addition, the spectra of **NA-3** and **NA-4** contained methoxy group in their structure, exhibited a longer absorption region than did the methoxy-free of **NA-2**. Furthermore, the **NA-3** compound exhibited a stronger red-shift than did the **NA-4** due to the better conjugation characteristics of *o*-methoxy than the m-methoxy substitute.

The photoluminescence (PL) results of **NA-1–4** are shown in Figure 1b and Table 1. The emission peaks of **NA-1–4** were located at 371, 371, 382, and 383 nm. Comparing **NA-1** to **NA-2**, although **NA-2** had a longer conjugation length through the 1-naphthalene substitution, both of them exhibited similar maximum emission peaks. Furthermore, **NA-2** exhibited a smaller PL intensity owing to the larger twist angle between 1-naphthalene and oxime ester [44]. In addition, the **NA-3** exhibited a higher PL intensity compared with the **NA-4** analog. According to the literature [45], the high emitting intensity might decline the triplet yield while the radicals are assumed to be generated from the triplet state. However, singlet state cleavage and the influence of the molecular structure on the intersystem crossing efficiency from single state to triplet state cannot be ruled out.

### 3.2. Thermal Properties

Thermal properties of the synthesized naphthalene-based oxime esters (**NA-1–4**) were investigated by thermogravimetric analysis (TGA) and differential scanning calorimetry (DSC). The TGA curves are presented in Figure 2a and their decomposition temperatures at the weight loss of 5% (T_d_) are summarized in Table 1. The T_d_ values are ranging from 146 to 167 °C. All of them had relatively low T_d_ values due to the easy degradation of the oxime ester unit in its structure. In addition, the **NA-2** with naphthalene substituted at 1 position had a relative lower T_d_ value than did the **NA-1** analog (naphthalene substituted at 2 position). This might be due to the steric hindrance between 1-naphthalene and oxime ester. Furthermore, the thermal stability of the **NA-2–4** compounds had the following order: **NA-4** > **NA-2** > **NA-3**, perhaps owing to the combination of the steric hindrance effect of 1-naphthalene unit and the methoxy substitution [44,46].

DSC confirmed that all the compounds had melting points and they exhibited the Tm values ranging from 80 to 111 °C (Figure 2b, Table 1). The melting point of the **NA-1–4** compounds had the following order: **NA-1** (111 °C) > **NA-4** (103 °C) > **NA-2** (88 °C) > **NA-3** (80 °C). The trend is similar with the thermal decomposition temperature mentioned above.

### 3.3. Electrochemical, Photolysis, and ESR Results

CV was employed to realize the electrochemical properties of the oxime esters. Figure 3 presents the oxidation and reduction potentials of the compounds under DCM solution. In addition, their related energy levels are summarized in Table 1. The oxidation potential of the compounds (Figure 3a) had the following order: **NA-1** (1.64 V) > **NA-2** (1.62 V) > **NA-3** (1.39 V) > **NA-4** (1.35 V). As expected, **NA-3** and **NA-4** oxidized at a relative lower oxidation potential because of the presence of an electron-donating methoxy substitution in its naphthalene unit [47]. Furthermore, the **NA-1** was oxidized at the highest potential due to the presence of the 2-naphthalene accompanied by methoxy-free in its structure. Furthermore, the reduction potentials of **NA-1–4** were investigated. The related curves are displayed in Figure 3b. The reduction potentials were ranging from -1.13 to -1.25 V. Generally, the LUMO orbitals for all the compounds are mainly composed of oxime ester moiety [18]. Among all, **NA-1** showed the lowest reduction potential as it contained 2-naphthalene substituted in its structure.

Steady state photolysis experiments of the selected **NA-1** and **NA-2** sample were carried out upon a Philips 16 W lamp (type Actinic BL; λ = 365 nm) (Figure 4). Within 2 min of light irradiation (λ = 365 nm), the selected compounds showed insignificant photolysis behavior because the **NA-1** and **NA-2** samples had poor absorption at 365 nm. However, another light irradiation source, which had a different light intensity region, might exhibit different photolysis characteristics.

We conducted radical concentration under UV irradiation (250 to 450 nm). The radicals generated by the selected **NA-2** and **NA-3** samples were conducted through ESR instruments. The related experimental ESR spectra are shown in Figure 5. Clearly, the spectral intensity of the **NA-3** compound is higher than that of the **NA-2** compound. Thus, the radical concentration of **NA-3** is higher and is expected to have better photoreactivity than the analogous **NA-2**-based one.

### 3.4. Photopolymerization Ability

The photopolymerizations of TMPTA monomer initiated by naphthalene-based oxime esters (**NA-1–4**) were conducted. First, no photopolymerization occurred without the light irradiation. Furthermore, the initial free radicals of the oxime esters are from the N–O bond cleavage process after light irradiation. Then, the corresponding naphthalene-based iminyl radical and methyl radical could be obtained for subsequent photopolymerization (Figure 2).

Thus, a photoinitiating system was prepared and exposed under the UV lamp and their photoreactivity results are shown in Figure 6 and Table 2. The **NA-1–4-** based formulations exhibited the maximum heat flow values of 520, 96, 347, and 149 mW mg^−1^, the maximum polymerization rates of 3.89, 0.74, 2.67, and 1.16 s^−1^, the time at maximum heat flow of 21, 44, 22, and 29 s, and the final conversions of 46, 33, 46, and 37%, respectively. The **NA-1**-based photoinitiating system exhibited the highest double bond conversion efficiency, maximum heat flow value, and the shortest time at maximum heat flow. This result might have been obtained because the **NA-1** had the highest molar extinction coefficient in the exposed region (250–450 nm). Comparing **NA-1** to the **NA-2** analogs, the **NA-2**-based formulation showed a poorer photopolymerization rate and its formulation was also the worst one. Furthermore, the photoreactivity was affected by the methoxy substituent, which decreased in the following order: **NA-3** > **NA-4** > **NA-2**. Clearly, the methoxy-free of **NA-2** and its formulation produced poor photoreactivity in contrast to other methoxy-substituted-based formulations. In addition, the **NA-3**-based formulation showed a higher photoreactivity, possibly due to the boarder absorption characteristic and higher radical concentration (see Figure 5).

Additionally, polymerization under LED light at 405 nm was also conducted; the results are shown in Figure 7 and Table 3. Interestingly, the trend of the photoreactivity is not consistent with the results obtained under UV light. Except for the **NA-2**-based formulation which does not react, the final degree of double-bond conversion decreased in the following order: **NA-3** (41%) > **NA-4** (24%) > **NA-1** (18%). The **NA-3**-based formulation exhibited a higher photoreactivity than did others because the **NA-3** had a rather red-shifted absorption region. The obtained results indicate that the **NA-3** oxime ester is the most suitable candidate for free radical polymerization under long or visible-light irradiation.

## 4. Conclusions

In this work, four oxime esters containing naphthalene, methoxy substitutions were prepared as Type I photoinitiators. Interestingly, the 1-naphthalene-substituted oxime ester (**NA-2**) exhibits better light harvesting but poorer photoreactivity compared with the 2-naphthalene-substituted one (**NA-1**). After incorporating the methoxy group into the 1-naphthalene unit (**NA-3** and **NA-4**), their photoractivities were changed significantly. Markedly, the *o*-methoxy substitute of the 1-naphthalene-based oxime ester (**NA-3**) not only exhibits a red-shifted absorption character, but shows the best photoinitiating performance upon UV and LED@405 nm light irradiations. This work paves the way for the design of new naphthalene-based oxime esters and their sufficient photopolymerization ability.

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
