# Peer review of "Naphthalene-Based Oxime Esters as Type I Photoinitiators for Free Radical Photopolymerization"

_polymers, 2022, doi:10.3390/polym14235261_

Round 1

Reviewer 1 Report

Chen, Lalevée and coworkers reported a very interesting work about a series of naphthalene-based oxime ester for photopolymerization.  This work is well organized and performed. However, there still some flaws in the manuscript, including references, writing, and experiments. Therefore, a  revision must be made before I can reconsider it for publication.

1. In the introduction, the authors said “photopolymerization generally classified into two pathway, (1) a free radical polymerization (FRP) [9] and (2) 35 a cationic polymerization (CP)”. When authors saying “free radical polymerization”, are you referring to the non-living radical polymerization? How about photoinduced controlled polymerization?  If the authors want to use radical polymerization to represent both living and nonliving polymerization, then in the following paragraph, the authors should not say “there are two types for FRP system that including (1) a Type I (one-component photoinitiator) and (2) a Type II photoinitiaing  system (two-components photoinitiator requiring the hydrogen acceptor and hydrogen donor package).” because the controlled radical polymerization used photocatalysts instead of photoinitiators.

2. Following question 1, the following recent examples of photopolymerization must be cited (https://doi.org/10.1039/C9PY01604J;  https://doi.org/10.1021/jacs.7b06413 https://doi.org/10.1021/acsmacrolett.0c00232)

3. Some typos. For example, “two pathway” ; should be “two pathways”. Please check the whole manuscript again.

4. The figure quality of Figure S1-S4 is low and need to be replaced with high quality figures. Current ones are really hard to read.

5. If it is possible, can authors also provide the CNMR for NA-1-4?

6. The photopolymerization mechanism using NA photoinitiators should be added in either the main text or SI, along with some discussion.

7. Control experiments without irradiation need to be added.

Author Response

  1. In the introduction, the authors said “photopolymerization generally classified into two pathway, (1) a free radical polymerization (FRP) [9] and (2) 35 a cationic polymerization (CP)”. When authors saying “free radical polymerization”, are you referring to the non-living radical polymerization? How about photoinduced controlled polymerization? If the authors want to use radical polymerization to represent both living and nonliving polymerization, then in the following paragraph, the authors should not say “there are two types for FRP system that including (1) a Type I (one-component photoinitiator) and (2) a Type II photoinitiaing  system (two-components photoinitiator requiring the hydrogen acceptor and hydrogen donor package).” because the controlled radical polymerization used photocatalysts instead of photoinitiators.

Thank you for your suggestions. We have made the modification of introduction part. In addition, the references mentioned below are also cited in the text.

  1. Following question 1, the following recent examples of photopolymerization must be cited (https://doi.org/10.1039/C9PY01604J; https://doi.org/10.1021/jacs.7b06413 https://doi.org/10.1021/acsmacrolett.0c00232)

Thank you for your suggestions. The references had been cited in refs 6-8.

  1. Some typos. For example, “two pathway” ; should be “two pathways”. Please check the whole manuscript again.

We thank the reviewer for her/his careful reading! The mistake has been corrected. In addition, we checked whole MS again.

  1. The figure quality of Figure S1-S4 is low and need to be replaced with high quality figures. Current ones are really hard to read.

Thank you for your suggestions. We had revised the figure quality of Figures S1-S4.

  1. If it is possible, can authors also provide the CNMR for NA-1-4?

Thank you for your suggestions. However, we didn’t have this data in our hands. We believed that all the compounds are synthesized successfully by using 1H NMR, high-resolution mass spectroscopy (HRMS) and elemental analysis (EA). We hope the reviewer can agree the publication of this paper without these data.

  1. The photopolymerization mechanism using NA photoinitiators should be added in either the main text or SI, along with some discussion.

Thank you for your suggestions. We had added the Scheme 2 and description in the text.

  1. Control experiments without irradiation need to be added.

Thank you for your suggestions. No photopolymerization was occurred without the light irradiation. We also added this description in the text.

Reviewer 2 Report

1. The title is not correct as photoinitiators generally don’t polymerize. Instead, they are used to initiate photopolymerization.  

2. NMR spectra are not clear. The authors should provide clear information using a software.

3. There are too many grammar mistakes that must be corrected.

4. Can the authors explain why the conversion is so low as 41%~46% when employing the Type I photoinitiator? The results seem implausible.

5. The key advantage of photopolymerization compared to thermal polymerization is to implement spatiotemporal control over the reaction, which is the basis of advanced manufacturing such as 3D printing and holographic patterning (Adv. Sci. 2022, 9 (10), 2105903; Nat. Commun. 2021, 12 (1), 2873; Angew. Chem., Int. Ed. 2020, 59 (25), 10066-10072; Acta Polym. Sin. 2022, 53 (7), 722-736; Macromolecules 2014, 47 (7), 2306-2315; Sci. China Mater. 2019, 62 (12), 1921-1933). However, such important characteristic is missing in the this text, making it difficult to attract broad attention.

6. Can the authors explain how does a Type I photoinitiator prevent the aggregation of unimolecular photoinitiator?

7. How does CO2 solve the polymerization inhibition by oxygen?

8. What do you mean when talking about “quantum yield from S1 ~ 20 %)”?

9. Equations used to for calculating the double conversion and polymerization rate are incorrect. The molecular weight and mass of the monomer should be considered when conducting P-DSC measurements (J. Am. Chem. Soc. 2014, 136 (25), 8855-8858; RSC Adv. 2014, 4 (9), 4420-4426).

10. Scheme to show the chemical synthesis should be provided for clarity.

11. How to understand the lowest reduction potential of NA-1 as the potential shown in Fig. 4b is the highest.

12. When comparing the EPR intensity, number should be provided in the Y axis of figure 6.

13. When discussing the light harvesting and photoinitiating performance, there is significant contradictory in the abstract and conclusion.  

Author Response

  1. The title is not correct as photoinitiators generally don’t polymerize. Instead, they are used to initiate photopolymerization.

Thank you for your suggestions. We had changed the title as “Naphthalene-based oxime esters as Type I photoinitiators for free radical photopolymerization”.

  1. NMR spectra are not clear. The authors should provide clear information using a software.

Thank you for your suggestions. We had revised the figure quality of Figures S1-S4.

  1. There are too many grammar mistakes that must be corrected.

Thank you for your suggestions. We had rechecked the whole MS.

  1. Can the authors explain why the conversion is so low as 41%~46% when employing the Type I photoinitiator? The results seem implausible.

Thank you for your suggestions. We are sorry for the description on the DC% calculation in the experiment part. We revised as DC = (ΔHt/n x ΔHotheor) × 100%. According to the reference, the theoretical reaction heat enthalpy of one acrylate for complete conversion was 86 kJ mol−1. Since the TMPTA used in this experiment has 3 acrylate, therefore we use 86*3 = 258 kJ mol−1 as the theoretical value. Such calculations may be risky due to the theoretical value ​​might not increase proportionally when increasing the number of acrylate. Therefore, this could be the reason for the low conversion.

  1. The key advantage of photopolymerization compared to thermal polymerization is to implement spatiotemporal control over the reaction, which is the basis of advanced manufacturing such as 3D printing and holographic patterning (Adv. Sci. 2022, 9 (10), 2105903; Nat. Commun. 2021, 12 (1), 2873; Angew. Chem., Int. Ed. 2020, 59 (25), 10066-10072; Acta Polym. Sin. 2022, 53 (7), 722-736; Macromolecules 2014, 47 (7), 2306-2315; Sci. China Mater. 2019, 62 (12), 1921-1933). However, such important characteristic is missing in the this text, making it difficult to attract broad attention.

Thank you for your suggestions. We had added the description and references in the introduction part.

  1. Can the authors explain how does a Type I photoinitiator prevent the aggregation of unimolecular photoinitiator?

Thank you for your suggestions. We had added the description in the introduction part. We changed as “but avoid the phenomenon of uneven mixability if a two-components system is used”.

  1. How does CO2 solve the polymerization inhibition by oxygen?

Thank you for your suggestions. We had added the description and revised the mechanism in introduction part.

  1. What do you mean when talking about “quantum yield from S1 ~ 20 %)”?

Thank you for your suggestions. We clearly specify as " singlet state quantum yield ~20 %".

  1. Equations used to for calculating the double conversion and polymerization rate are incorrect. The molecular weight and mass of the monomer should be considered when conducting P-DSC measurements (J. Am. Chem. Soc. 2014, 136 (25), 8855-8858; RSC Adv. 2014, 4 (9), 4420-4426).

Thank you for your comment and suggestion. We are really appreciated for your information. We also agree with the calculation equation that you mentioned. However, we would worry about the theoretical heat of the acrylate (86 mol/kJ) will change depend on the different light intensity and irradiation wavelength. Therefore, in this system, we use the simplest calculation method for the theoretical value (n*86, n=3) to understand the related conversion efficiency.

  1. Scheme to show the chemical synthesis should be provided for clarity.

Thank you for your suggestions. We had revised the Scheme 1 with the chemical synthesis procedure.

  1. How to understand the lowest reduction potential of NA-1 as the potential shown in Fig. 4b is the highest.

Thank you for your suggestions. We are sorry put the wrong order for the CV curves. All the oxidation and reduction potentials were studied from the transition point of the related spectra (Figure 3a and 3b)

  1. When comparing the EPR intensity, number should be provided in the Y axis of figure 6. Thank you for your suggestions. We had added the number of Y axis for the ESR spectrum.

  1. When discussing the light harvesting and photoinitiating performance, there is significant contradictory in the abstract and conclusion.

Thank you for your suggestions. We had revised the description in the abstract and conclusion part.

Reviewer 3 Report

The manuscript 'Naphthalene-based oxime esters as Type I photoinitiators for free radical photopolymerization' describes design and synthesis of a series of naphthalene-based oxime esters and the study of these compounds as photoinitiators.

However, I have a number of significant remarks to this manuscript.

1. The Abstract have been prepared in complete disregard of the Instructions for Authors.

I quote: 'The abstract should be a single paragraph and should follow the style of structured abstracts, but without headings: 1) Background: Place the question addressed in a broad context and highlight the purpose of the study; 2) Methods: Describe briefly the main methods or treatments applied. Include any relevant preregistration numbers, and species and strains of any animals used. 3) Results: Summarize the article's main findings; and 4) Conclusion: Indicate the main conclusions or interpretations.'

2. In Section 2.2, the authors have described the synthesis of new (?) organic compounds. The common rules for identification and description of new compounds require registration of 13C NMR spectra and presentation of the data obtained. 13C NMR spectra also should be presented in the SI file.

Minor remarks:

Line 33 – abbreviation 'VOCs' is introduced without subsequent use

Lines 49-51 – pleas clear the sentence

Line 53 – are currently

Line 55 – 'was easily cleavage' –correct this

Line 65 – is it a Scheme? The Scheme number and description are needed

Line 83 – the names of the chemicals' suppliers are needed (company name, city, country), see Instructions for Authors

Line 90 and below – extensive language editing is needed

Line 93 and below – the use of subsection numbering (2.2.1 etc) is needed

Line 146 – 'The proton-nuclear magnetic resonance (1H NMR) spectra of the samples were dissolved' – please watch your language

Line 185 – please correct according to the manuscript template for equations

Line 208 – design of the Scheme 1 is bad and need to be reworked

Line 313 – Scheme 2, similarly

etc etc, the text needs careful editing.

In the present form the manuscript contains very large amount of mistakes and inaccuracies. Careful pre-editing is needed. In summary, in the present form the article is not suitable for publication in Polymers journal, and should be reworked thoroughly. Moreover, I'm note sure that this article is suitable for Polymers since it doesn't have 'macromolecule' content. I would recommend publication of this work in Molecules journal, of course, after more careful preparation of the manuscript.

Author Response

  1. The Abstract have been prepared in complete disregard of the Instructions for Authors.I quote: 'The abstract should be a single paragraph and should follow the style of structured abstracts, but without headings: 1) Background: Place the question addressed in a broad context and highlight the purpose of the study; 2) Methods: Describe briefly the main methods or treatments applied. Include any relevant preregistration numbers, and species and strains of any animals used. 3) Results: Summarize the article's main findings; and 4) Conclusion: Indicate the main conclusions or interpretations.'

Thank you for your suggestions. We try to revised to meet the article instructions. In addition, the article that we had including (1) abstract, (2) Introduction, (3) Experimental, (4) Results and discussion and (5)Conclusions. We believed we’ve covered all parts, and there are not much different from the content of recent journals published in Polymers [ref A, B, C].

[ref A] Sun, X.; Wang, J.; Fu, Q.; Zhang, Q.; Xu, R. Synthesis of a Novel Bifunctional Epoxy Double-Decker Silsesquioxane: Improvement of the Thermal Stability and Dielectric Properties of Polybenzoxazine. Polymers 2022, 14, 5154.

[ref B] Grommes, D.; Schenk, M. R.; Bruch, O.; Reith, D. Initial Crystallization Effects in Coarse-Grained Polyethylene Systems After Uni- and Biaxial Stretching in Blow-Molding Cooling Scenarios. Polymers 2022, 14, 5144.

[ref C] Fredi, G.; Favaro, M.; Da Ros, D.; Pegoretti, A.; Dorigato, A. Thermotropic Optical Response of Silicone–Paraffin Flexible Blends Polymers 2022, 14, 5117.

  1. In Section 2.2, the authors have described the synthesis of new (?) organic compounds. The common rules for identification and description of new compounds require registration of 13C NMR spectra and presentation of the data obtained. 13C NMR spectra also should be presented in the SI file.

Thank you for your suggestions. The 13C NMR spectra have been added in supporting information and experimental part.

  1. Minor remarks

Thank you for your suggestions. We have made the modification of the whole manuscript. We hope the reviewer can satisfied of this paper with the revised draft.

  1. In the present form the manuscript contains very large amount of mistakes and inaccuracies. Careful pre-editing is needed. In summary, in the present form the article is not suitable for publication in Polymersjournal, and should be reworked thoroughly. Moreover, I'm note sure that this article is suitable for Polymers since it doesn't have 'macromolecule' content. I would recommend publication of this work in Molecules journal, of course, after more careful preparation of the manuscript.

Thank you for your suggestions. In this article, in addition to discussing the characteristics of the novel photoinitiators, we also discuss the photopolymerization characteristics of the photoinitiators under different light sources. We think this manuscript may add some new information to the area of photopolymerizations. We sincerely hope the reviewer’s consideration of its publication.

Reviewer 4 Report

The revision is suitable for direct publication 

Author Response

We appreciated the reviewer's postive comment.

Round 2

Reviewer 1 Report

The authors revised the manuscript with care

The current one is very good and should be published as it is

Author Response

(The authors gave the same response as above.)
